# No evidence for a mixing benefit—A registered report of voluntary dialect switching

Mathieu Declerck[1], Neil W. Kirk [2]*

**1** Linguistics and Literary Studies, Vrije Universiteit Brussel, Brussels, Belgium, **2** Division of Psychology & Forensice Sciences, Abertay University, Dundee, Scotland, United Kingdom

* n.kirk@abertay.ac.uk

**Data Availability Statement:** All data files are available from the Open Science Framework at

## Abstract

Previous language production research with bidialectals has provided evidence for similar language control processes as during bilingual language production. In the current study, we aimed to further investigate this claim by examining bidialectals with a voluntary language switching paradigm. Research with bilinguals performing the voluntary language switching paradigm has consistently shown two effects. First, the cost of switching languages, relative to staying in the same language, is similar across the two languages. The second effect is more uniquely connected to voluntary language switching, namely a benefit when performing in mixed language blocks relative to single language blocks, which has been connected to proactive language control. While the bidialectals in this study also showed symmetrical switch costs, no mixing effect was observed. These results could be taken as evidence that bidialectal and bilingual language control are not entirely similar.

## Introduction

Previous research has indicated that when bidialectals (i.e., speakers of a regional dialect that are also fluent in a standard language variety) produce language, both the standard language and dialect are activated (e.g., [1, 2]), which is assumed to lead to competition among both language varieties. Similar to bilinguals (for reviews, see [3, 4]), a language control process is assumed to be implemented to deal with the competition between language varieties in bidialectals [1–2, 5, 6]. Some studies have suggested that the language control process implemented by bidialectal speakers of closely related language varieties is similar to the bilingual language control process [1, 2, 5]. In the current study, we set out to further investigate the language control process implemented by bidialectals and its relation to the bilingual language control process by letting bidialectals perform in a voluntary language switching paradigm.

Voluntary language switching [7–16] usually requires participants to name pictures in one of two languages. Unlike other variants of the language switching paradigm (for a review, see [4]), which indicate the language that should be used for each stimulus through cues (e.g., differently colored frames around the stimuli; [17]), alternating languages (e.g., AABBAABB, with A and B referring to trials in different languages; [18]), or written words (e.g., [19]), voluntary language switching allows the participants to choose the language on each trial.

https://osf.io/wmr3d/ (DOI: 10.17605/OSF.IO/WMR3D).

**Funding:** This research was funded by a Carnegie Trust for the Universities of Scotland Research Incentive Grant (https://www.carnegie-trust.org/) awarded to NWK (RIG009864). The funders did not have a role in study design, data collection and analysis, decision to publish, or preparation of the manuscript.

**Competing interests:** The authors have declared that no competing interests exist.

Similar to other variants of language switching (e.g., [17, 20–23]), voluntary language switching with bilinguals generally results in a cost when switching languages, relative to staying in the same language across trials ([8–16]; however, see [7]). Indicative of the voluntary language switching paradigm with bilinguals is that these switch costs are similar across languages (i.e., symmetrical switch costs [8–14]). This is in contrast with other variants of the language switching paradigm, where asymmetrical switch costs, which entails larger switch costs in the first language (L1) than in the second language (L2) and is typically used as a measure of inhibitory control (e.g., [17, 24–26] for a review, see [27]), are relatively often found. According to Gollan and Ferreira [11], the absence of asymmetrical switch costs with the voluntary language switching paradigm is because this paradigm allows bilinguals to name "easier" words in their L2. This should result in a more similar L1 and L2 activation level for the produced words regardless of language proficiency, and thus might lead to symmetrical switch costs.

Another measure of language control are mixing costs (for a review, see [28]). Mixing costs entail worse performance in repetition trials in mixed language blocks relative to performance in single language blocks. This measure has been explained with control processes that are implemented in anticipation of any upcoming cross-language competition (i.e., proactive language control; e.g., [26]) and the mental cost to maintain and monitor two languages (e.g., [8]). Mixing costs are a highly stable effect in all variants of language switching (e.g., [20, 25, 26, 29–31]), with the exception of voluntary language switching. Voluntary language switching studies actually tend to show a mixing benefit (i.e., worse performance in single language blocks relative to performance in repetition trials in mixed language blocks) in one [11, 12] or both languages [8, 9, 13, 14]. The mixing benefit in L2 observed by Gollan and Ferreira [11] was explained by assuming that only "easier" words are produced in L2 in the voluntary language switching paradigm. Based on this explanation, one would expect that the mixing benefit was only observed for words that were consistently named in L2, but that was not the case [11]. An alternative explanation that can account for a mixing benefit across both languages comes from de Bruin et al. [8]: Production in a single language block requires substantial proactive inhibition of the non-target language [32, 33]. When producing in a mixed language block with voluntary language switching, bilinguals do not require substantial proactive control processes to guide language production, since participants can choose which language to use on any given trial. So, because the implementation of more proactive control processes during single than mixed language blocks is more taxing, better performance is expected in the latter block type when using a voluntary language switching paradigm. From this overview, it appears that voluntary vs. involuntary language switching has a large impact on measures of the bilingual language control process.

In the current study, we set out to investigate if this is also the case for bidialectals by asking bidialectals to perform in a voluntary language switching paradigm. The few studies that investigated control processes with bidialectals and bilinguals provide evidence for a shared language control process. Similar to bilinguals, bidialectals show a cost to switching between language varieties, relative to staying in the same language variety across languages [1, 2, 5, 6]. Kirk and colleagues [5], for instance, let bidialectals (English-Orcadian) name pictures during a language switching task, where the language variety on each trial was indicated by differently colored frames corresponding to each language variety (cf. involuntary language switching). The results showed that switching between language varieties results in worse performance than staying in the same language variety across trials. Similar to bilinguals (e.g., [17]), these bidialectals showed asymmetrical switch costs, with larger switch costs in their more dominant language variety (dialect) than in their less dominant language variety (standard language; see also [2]). Asymmetrical switch costs were even found with new bidialectals (English-

Dundonian), whereas more fluent bidialectals showed symmetrical switch costs [1]. The latter pattern is similar to that observed with second language learners and highly proficient bilinguals, respectively ([21, 34]). Finally, Kirk and colleagues also showed worse performance in repetition trials in mixed language blocks than in single language blocks [5]. So, along the lines of prior bilingual studies (e.g., [26, 29–31]), mixing costs can be observed with bidialectals during involuntary language switching.

The similarities of bidialectals and bilinguals in previous studies that relied on involuntary language switching seem to indicate that similar language control processes are implemented by these two groups during language production. While it might seem obvious that bilinguals and bidialectals rely on the same language control processes, previous related research indicates that is not necessarily the case. For instance, control processes used within the same language have been shown to be different to control used between languages [35]. Even more damning for the assumption that similar control processes are used throughout language processing is that some studies found evidence that different language pairs do not necessarily converge when it comes to language control [36]. A similar discrepancy in language control has been observed across modalities within the same bilinguals (e.g., [37]). These studies provide evidence against the claim that language control is domain general [17, 24], as the control processes within a domain (i.e., language processing) are sometimes even different.

There have also been attempts to objectively distinguish languages from dialects (and thus bilinguals from bidialectals) on a cognitive level using the picture word interference paradigm, which initially suggested dialect items were processed as within-language competitors, akin to synonyms [38]. However, more recent evidence has challenged some of these findings [39, 40]. Thus, the extent to which bidialectals are similar to bilinguals is still unclear and can have theoretical and methodological implications for research comparing bilinguals and monolinguals. For example, research suggesting that there is a general executive control advantage for bilinguals over monolinguals as a result of the regular engagement of language control mechanisms [41], could be invalidated by the presence of bidialectal speakers who also use these mechanisms, but who are erroneously categorized as monolingual [42].

To further investigate the issue of whether there are similar language control processes used in bilingual and bidialectal language production, we set out to examine whether a similar pattern can be observed with bidialectals as with bilinguals in a voluntary language switching paradigm. In the current study, we relied on Scottish speakers of a specific type of Scots as the dialect of interest. Although Scots is recognized by the European Charter for Regional or Minority Languages as a separate minority language (from English), it is generally not given this status, with many facing ridicule for suggesting that Scots and English are separate languages [43]. Even a majority of speakers themselves do not hold this view, with one Scottish Government [44] report demonstrating that 65% of respondents consider their use of Scots as "just a way of speaking". Consequently, these speakers are likely to identify as monolingual rather than bilingual—or even bidialectal–especially if language background measures are not sensitive to the existence of non-standard varieties (see, [1, 5]).

More specifically, we will test speakers of Dundonian Scots and (Scottish) Standard English. Dundonian is an urban dialect used in and around the Scottish city of Dundee. Like other urban Scots dialects, it exists as a lower status variety in a diglossic situation with (Scottish) Standard English as the prestige variety [45]. Whereas Dundonian Scots overlaps substantially with its corresponding standard language, there are several notable differences. First, it is characterized by phonetic differences relative to the standard language, such as vowel differences (e.g., Standard English 'pie' /paɪ/ vs. Dundonian 'peh' /pɛ/) and monophthongisation (e.g., Standard English 'mouse' /maʊs/ vs. Dundonian 'moose' /mu:s/). Second, and most important for the current study, there are also words entirely different in Dundonian than its

corresponding translation equivalent in Standard English (e.g., "crying" in Standard English would be "greetin" in Dundonian).

If bidialectals and bilinguals rely on similar language control processes, we expect to observe symmetrical switch costs in the voluntary language switching paradigm with the English-Dundonian bidialectals, similar to the pattern observed with bilinguals. This finding might simply indicate that bidialectals tend to produce "easier" words in the less proficient language [11]. Observing a mixing benefit with English-Dundonian bidialectals in a voluntary language switching paradigm would be a more unique finding, as this effect has only reliably been observed with bilinguals in a voluntary language switching paradigm. A mixing benefit would indicate that bidialectals implement proactive language control in single language blocks, whereas this is less the case in mixed language blocks [8].

## Method

### Participants

A power analysis on the mixing benefit was run to determine the required number of participants and trials. We chose the mixing benefit, since this effect seems more uniquely connected to voluntary language switching. Along the approach suggested by Brysbaert and Stevens [46], we ran 200 (Monte Carlo) simulations with the simr package [47] on the voluntary language switching data of Jevtović et al. [14], who tested 40 bilinguals with 20 distinct stimuli in 80 trials in single language blocks and 180 trials in voluntary language switching blocks for each participant. The results showed that the setup of Jevtović et al. [14] had a 99.5% chance of showing a mixing benefit. In the current study, we relied on the same number of stimuli as Jevtović and colleagues. Furthermore, we relied on a similar number of participants and trials. The number of voluntary language switching trials per participant was slightly decreased relative to Jevtović et al. (from 180 to 160 trials per participant). That way, we had a similar number of repetition trials in the voluntary language switching blocks and overall trials in the single language blocks. To make sure that we had at least the same number of voluntary language switching block trials across participants as Jevtović and colleagues, we increased the number of participants from 40 to 46.

In line with this target, we collected useable data from 46 bidialectal speakers of (Scottish) Standard English and Dundonian Scots (16 identified as men, 29 as women, and 1 as non-binary), who were recruited through a local media campaign and word of mouth. An additional two participants completed the study but did not produce useable data–one had an uploading error resulting in no audio recordings being stored and another was excluded at the accuracy/variety coding stage due to eating throughout the task, which interfered with reaction time extraction.

Using a similar questionnaire for the bidialectals as in Kirk et al. [1] and Declerck et al. [48], these speakers reported using Dundonian Scots around 26% of the time. Following the experiment, the participants also completed an English vocabulary test on the basis of a lexical decision task (i.e., LexTale [49] and a 10-item Dundonian Scots test where they had to select the correct definition of a word from four options). This information and a summary of demographic information is shown in Table 1. The study received approval from Abertay University's research ethics committee (EMS4259).

### Materials and task

Along the lines of Jevtović et al. [14], 20 pictures were presented to the bidialectal participants. These pictures corresponded to non-cognate names between Standard English (average number of syllables across the English names: 1.55; average Zipf frequency of the English names:

**Table 1. Overview of participants' demographic information (SD in brackets).**

| N | 46 |
|---|---|
| Age (years) | 38.3 (10.7) |
| Dundonian use (Current %) | 25.9 (13.6) |
| Dundonian use (Childhood %) | 27.4 (20.4) |
| Self-rated Dundonian comprehension* | 6.4 (0.8) |
| Self-rated English comprehension* | 7 (0) |
| Self-rated Dundonian production* | 5.7 (1.3) |
| Self-rated English production* | 6.9 (0.2) |
| Dundonian Word Test (%) | 87.8 (12.3) |
| English LexTale (%) | 91.4 (10.7) |
| Dundonian Switch Rate (%)** | 22.9 (2.9) |
| Dundonian Repetition Rate (%)** | 33.4 (11.9) |
| English Switch Rate (%)** | 22.4 (3.3) |
| English Repetition Rate (%)** | 21.2 (12.3) |

Note.* Self-rated scores are on a scale of 1 (low proficiency) to 7 (high proficiency).

** Switch and Repetition rates are based on percentage of total number of valid coded trials in voluntary language switching blocks, before reaction time clean up. The variety name refers to whether the trial was named in Dundonian or English. Switch refers to having used the other variety in the previous trial and Repetition refers to having remained within that variety from the previous trial.

4.24; [50]) and Dundonian (average number of syllables across the Dundonian names: 1.60; Zipf frequency was not calculated for Dundonian names as no database with word frequency exists for the Dundonian dialect. See S1 Table for the stimulus list). Each picture was presented twice, in non-consecutive trials, throughout each block.

## Procedure

The picture naming study was presented online on the Gorilla platform [51] and is publicly available as Open Materials (https://app.gorilla.sc/openmaterials/341909). After providing informed consent, participants performed a microphone check, in which they named a sentence and then listened to their own recording. A familiarization block followed the microphone check, in which all 20 pictures were presented together with the corresponding names in Standard English and Dundonian. Because there is no standardized written form of the Dundonian dialect, participants had the option to listen to a recording of the Dundonian word spoken by a local speaker. The familiarization phase was followed by the actual experiment, which consisted of two single language blocks of 40 trials each and four voluntary language switching blocks of 40 trials each. There are several ways to present the order of single language blocks and mixed language blocks to obtain mixing costs or a mixing benefit [28]. We opted for a setup in which participants first saw one single language block, followed by the four voluntary language switching blocks, and then again one single language block in the other language variety than the first single language block. The language variety of the single language blocks was counterbalanced across participants.

Prior to each of the three block types (i.e., English language block, Dundonian language block, and the voluntary language switching blocks), instructions were displayed pertinent for that block type, emphasizing speed and accuracy to name each picture. Moreover, in the single language blocks, the bidialectals were instructed to name each picture in the corresponding language throughout the block. Prior to the voluntary language switching blocks the following

sentences were presented (for similar instructions, see [8, 14]): "In the following section, you can name the pictures in either Standard English or Dundonian. You are free to choose which language variety to use for each picture. However, please do not use the same language variety throughout the whole task.". The instructions were followed by a short demonstration of the task before completing a short practice block consisting of 8 trials. Finally, the participants performed the experimental block(s).

Each trial started with a fixation cross in the middle of the screen. After 250 ms, the fixation cross was replaced by the stimulus for a maximum of 3000 ms. Each trial ended with a brief 50 ms blank screen before the onset of the next trial.

Following the main task, participants completed a Language Background Questionnaire, the English LexTale and a 10-item test of Dundonian Scots words (also available from: https://app.gorilla.sc/openmaterials/341909).

## Data analyses

The raw data and analyses scripts are available on the Open Science Framework (https://osf.io/wmr3d/).

The first trial in each voluntary language switching block was excluded from the reaction time (RT) analyses, since this is neither a switch nor repetition trial. Furthermore, error trials and trials immediately following an error were also excluded from the RT analyses. Trials with RTs faster than 150 ms, slower than 3000 ms, or three standard deviations above participant mean were also removed.

Similar to previous bilingual voluntary language switching studies (e.g., [9, 10]), we provided an overview of the mean switch rate of the participants in all conditions (see Table 1). Furthermore, two analyses were conducted on the RT data. The Switching analysis was conducted on the data of the voluntary language switching blocks and consisted of the factors Trial type (switch vs. repetition trials) and Language variety (Standard English vs. Dundonian). The Mixing analysis was conducted on the "pure" trials from the single language blocks and the repetition trials in the voluntary language switching blocks. The latter analysis consisted of Block type (repetition trials from voluntary language switching blocks vs. trials from single language block) and Language variety (Standard English vs. Dundonian).

The RTs were analyzed using linear mixed-effects regression modeling [52]. Both participants and items were considered random factors with all fixed effects and their interactions varying by all random factors [53]. No, convergence issues were registered, hence we did not need to rely on the buildmer package [54] to simplify the model. For all two-level factors we used effect coding (i.e., -0.5 and 0.5). Finally, $t$- and $z$-values larger or equal to 1.96 were deemed significant [55].

The error data were not analyzed because we did not observe enough errors for meaningful analysis (< 5%). However, we did report descriptive statistics below.

## Results

### Error rates

Trials were coded as errors if the wrong word was produced, if no utterance was produced, or if the utterances contained extraneous noises (e.g., "umm"). Additionally, in the Single language blocks, trials produced in the wrong variety were coded as errors (this was not relevant in the Voluntary language switching blocks as the participants were free to produce in either variety from trial to trial).

Overall, error rates reached 3.5% of valid trials, thus we did not conduct any error analyses in line with our threshold of 5% errors (or more). For a breakdown of the error rates, see Table 2.

**Table 2. Breakdown of percentage errors in each block type.**

| Single Language Block | | Voluntary Language Block |
|---|---|---|
| 4.97% | | 2.70% |
| Standard | Dialect | |
| 4.33% | 5.62% | |

## Reaction times

**Switching analysis.** The model yielded a main effect of Trial Type, with slower responses in switch (990.3 ms, SD = 289.2 ms) than in repetition trials (962.6 ms, SD = 268.9 ms; see Table 3 and Fig 1), indicating that voluntary language switching with bidialectals still produces switch costs. We also found a main effect of Variety with Standard English items (982.3 ms, 286.7 ms) being named slower on average than Dundonian Scots dialect items (969.5 ms, SD = 272.0 ms). There was no interaction between Trial Type and Variety, indicating a symmetrical switch cost pattern across language varieties.

**Mixing analysis.** The model yielded no main effect of Block Type, indicating similar reaction times for the repetition trials from the voluntary language switching blocks and the trials from the single language blocks, thus providing no indication of either a mixing benefit or cost. There was a main effect of Variety with Standard English items (989.7 ms, SD = 283.5 ms) being named slower than Dundonian Scots items (959.8 ms, SD = 267.4 ms; see Table 3 and Fig 2). There was no significant interaction between Block Type and Variety.

## Discussion

In the current study, we set out to examine whether a similar language control process is implemented by bilinguals and bidialectals by relying on voluntary language switching. Prior bilingual studies have shown that voluntary language switching is easier (at least in one language) than producing in a specific language (i.e., mixing benefit; e.g., [8, 9, 11–14]), an effect linked to proactive language control [8]. Additionally, bilingual voluntary language switching tends to result in symmetrical switch costs across the two languages [8–14]. We reasoned that similar patterns should be observed with bidialectals if bilinguals and bidialectals rely on a similar language control process. On the one hand, the results showed no mixing benefit with bidialectals. On the other hand, symmetrical switch costs were found during bidialectal voluntary language switching, similar to most voluntary language switching studies with bilinguals.

**Table 3. Parameter estimates and results of significance tests in mixed-effects models.**

| Fixed effects | β | SE | t | p |
|---|---|---|---|---|
| Switching Model: Reaction Times ~ Variety * TrialType + (Variety * TrialType\| Participant) + (Variety * TrialType\| Picture) | | | | |
| Intercept | 991.47 | 29.30 | 33.84 | < .001 |
| Variety (Standard vs Dialect) | -42.02 | 15.80 | -2.66 | .013 |
| Trial Type (Switch vs Repeat) | -24.64 | 7.88 | -3.13 | .003 |
| Trial Type * Variety | -1.16 | 12.93 | -0.09 | .929 |
| Mixing Model: Reaction Times ~ Variety * BlockType + (Variety * BlockType \| Participant) + (Variety* BlockType \| Picture) | | | | |
| Intercept | 979.90 | 27.43 | 35.72 | < .001 |
| Variety (Standard vs Dialect) | -54.69 | 18.07 | -3.03 | .005 |
| Block Type (Single vs Voluntary) | -3.14 | 13.05 | -0.24 | .811 |
| Block Type * Variety | 20.18 | 19.88 | 1.02 | .316 |

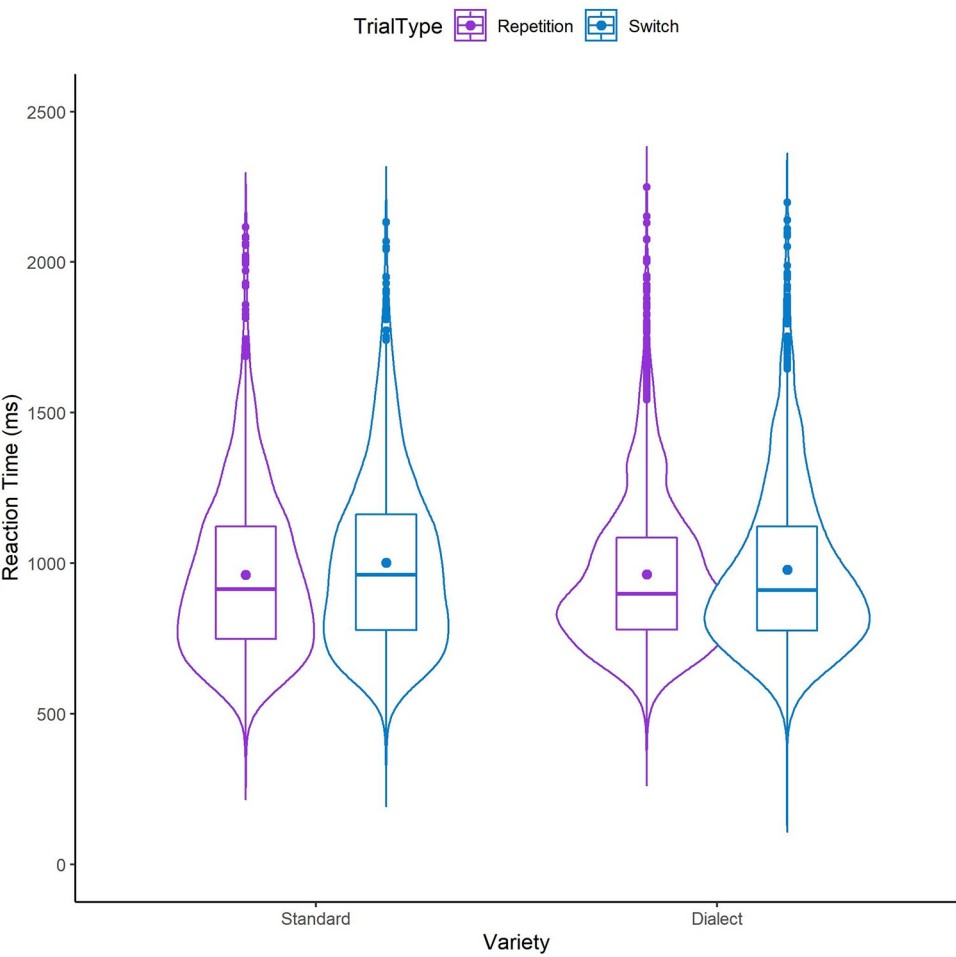

**Fig 1. Violin plot showing the distribution of reaction times across switch and repetition trials in the voluntary language switching block, indicative of switching costs for Standard English (Standard) and Dundonian Scots (Dialect).** The boxplot shows the interquartile range, the horizontal line represents the median, and the dot indicates the mean for each condition.

## Bidialectal vs. bilingual voluntary mixing effect

Regarding the unique mixing benefit patten (i.e., worse performance in single language blocks than in mixed language blocks, at least for one language) observed in bilingual studies that relied on voluntary language switching [8, 9, 11–14], no such pattern was observed in the current bidialectal study. More specifically, we observed no significant mixing effect. Since the mixing benefit with a bilingual voluntary language switching task is typically explained in terms of proactive language control [8], the lack of such a mixing benefit with bidialectals could be interpreted as there being no proactive language control, or at least no difference in proactive language control between single- and mixed-language contexts.

It is surprising that we observed no voluntary mixing benefit with bidialectals because prior bidialectal studies that investigated language control [1, 2, 5] typically showed quite some overlap with the findings of similar bilingual studies. Yet, it should be noted that several studies have shown that qualitative differences in language control can be observed across bilingual groups (e.g., [56, 57]). In a recent study by Declerck et al. [56], for instance, a different ERP pattern was observed when bimodal bilinguals (i.e., bilinguals proficient in a signed and

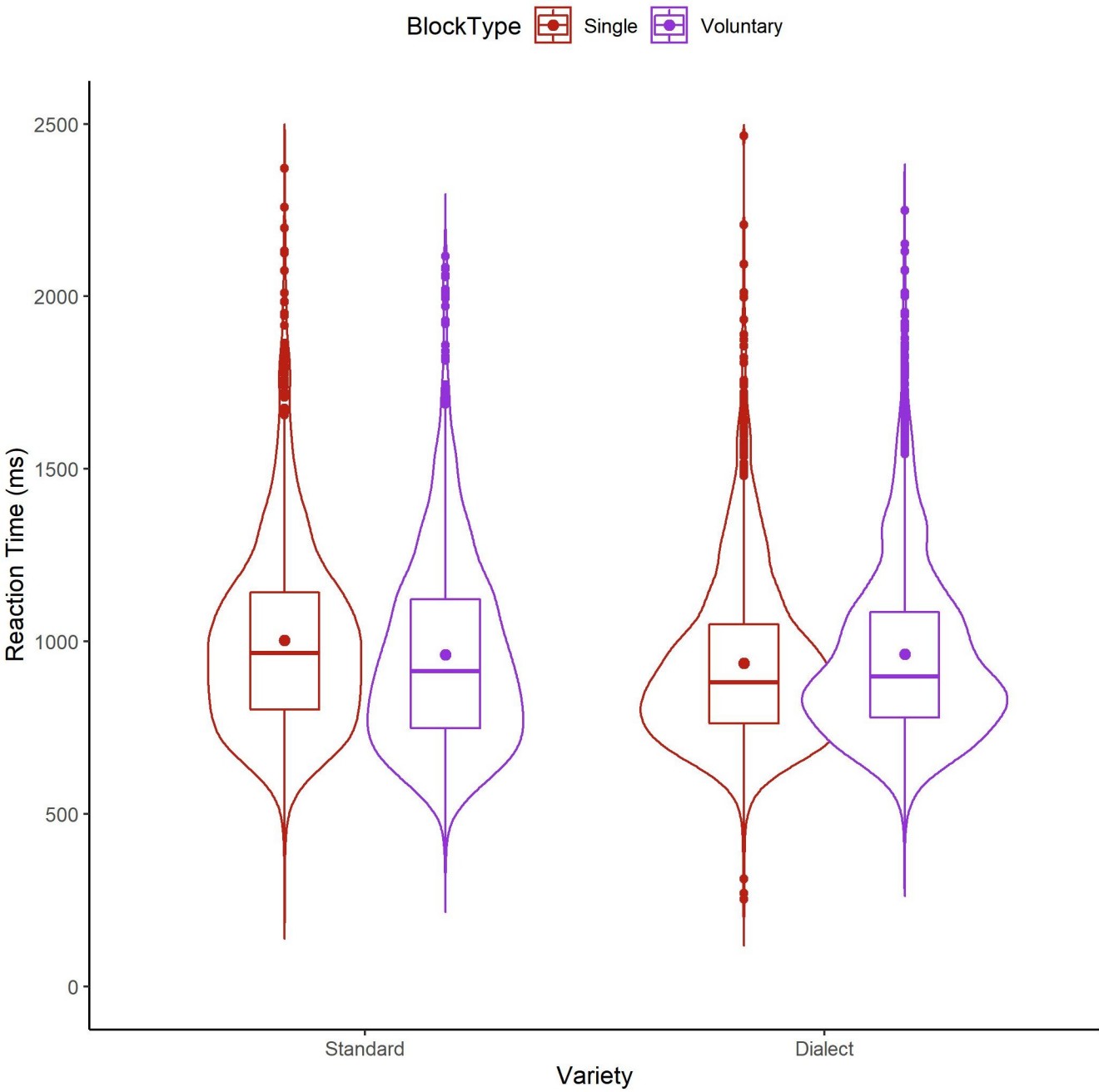

**Fig 2. Violin plot showing the distribution of reaction times across trials in the single language blocks and repetition trials in the voluntary language switching blocks, indicative of a mixing effect for Standard English (Standard) and Dundonian Scots (Dialect).** The boxplot shows the interquartile range, the horizontal line represents the median, and the dot indicates the mean for each condition.

spoken language) switched languages involuntarily relative to what is typically observed with unimodal bilinguals (i.e., bilinguals proficient in two spoken languages), thus showing a difference in language control between these two groups of bilinguals. Similarly, the results of the current study could be interpreted as bidialectals and bilinguals not relying on exactly the same language control processes. However, any language mixing difference observed here might also be due to differences between the specific languages used in prior bilingual

voluntary language switching studies (Catalan-Spanish, Spanish-Basque, and Spanish-English) and the language varieties used in the current study. So, more research based on different languages and language varieties should result in more conclusive evidence, especially since the few bilingual studies that investigated the voluntary mixing benefit relied on a small number of language combinations.

An alternative explanation is that the absence of a significant mixing benefit with bidialectals during voluntary language switching is due to the same underlying mechanism that results in a mixing benefit with bilinguals that perform a voluntary language switching task [8, 9, 11–14]. As we have indicated in the introduction, both bidialectals [5] and bilinguals [20, 25, 26, 29–31] show a mixing cost when involuntarily switching between languages. With bilinguals this turns into a mixing benefit, for at least one language, when switching languages voluntarily. This reversal of the mixing effect could be due to little to no language control being necessary in voluntary mixed language blocks, whereas some control processes are necessary in involuntary mixed language blocks. Based on the current study, no such mixing benefit pattern is found with bidialectals. However, relying on voluntary language switching does change the typical mixing cost effect observed with involuntary language switching into a non-significant effect. Therefore, not only does voluntary language switching impact the mixing effect of bilinguals, but also bidialectals, when compared to involuntary language switching. It might be that the underlying mechanism that turns a mixing cost into a mixing benefit due to bilingual voluntary language switching could also change the mixing cost pattern observed during bidialectal involuntary language switching to the point that this effect is not significant anymore (instead of becoming a mixing benefit with bilinguals) when relying on voluntary language switching. This would entail that there is no qualitative difference in (proactive) language control between bidialectals and bilinguals, but merely a quantitative difference.

## (A)symmetrical switch costs

Along the lines of previous involuntary language switching studies with bidialectals [1, 2, 5, 6], the present study showed that a switch cost pattern can be observed when bidialectals voluntarily switch between their dialect and the standard language. Similar to bilingual voluntary language switching studies [8–14], the size of the switch costs was similar for the dialect and the standard language. This could be taken as evidence that at least some language control processes implemented by bidialectals and bilinguals in a voluntary language switching context are similar.

Another interpretation would be that bilinguals and bidialectals have a tendency to produce easier words in their less proficient language during voluntary language switching [11]. This should lead to more similar activation levels across the two languages, at least for the used items, and thus could result in similar switch costs. Hence, it could be that that this pattern simply indicates that bilinguals and bidialectals tend to rely on the same strategy during voluntary language switching.

Furthermore, it should be taken into account that prior studies with a similar group of bidialectals performing involuntary language switching tasks also resulted in symmetrical switch costs [1, 48]. This is not to say that bidialectals never show asymmetrical switch costs when involuntarily switching languages (see [5]), but members of this particular group of bidialectals (i.e., Dundonian-English bidialectals) might generally be similar to highly proficient bilinguals (e.g., [21, 34]), which should have a smaller difference between the base activation of their two language varieties, and thus show similar switch costs across language varieties.

Along the same lines, it is still unclear what the boundary conditions are to observe asymmetrical switch costs (for reviews, see [4, 27, 58]). So, finding symmetrical switch costs might also be due to characteristics of this study other than voluntary language switching.

Another reason why not too much weight should be put upon finding a similar symmetrical switch cost pattern during voluntary language switching with bilinguals and bidialectals is that two very recent bilingual voluntary language switching studies observed asymmetrical switch costs [59, 60]. Hence, it would seem as if the symmetrical switch cost pattern in bilingual voluntary language switching is less stable than we originally thought at the start of this project. This makes it more difficult to make any comparisons between bilingual and bidialectal voluntary language switching results based on this effect.

## Conclusion

In sum, a different mixing pattern was observed in this Registered Report (see Declerck & Kirk [61] for the stage 1 Registered Report of the current study) with bidialectals when relying on voluntary language switching (i.e., no significant mixing effect) than what is generally observed with bilinguals that perform a voluntary language switching task (i.e., a mixing benefit in at least one language). While this could be explained with the notion that these two groups rely on, at least partly, different language control processes, it might just be due to a quantitative difference.

Regarding the symmetrical switch costs observed with bidialectals in a voluntary language switching task, which is similar to what has typically been found with bilinguals, it is prudent to be careful to interpret this pattern as evidence that bilinguals and bidialectals rely to some degree on the same language control processes, as other interpretations (cf. similar strategies, high proficiency in both language varieties, and/or specific experiment characteristics) are also valid.

## Supporting information

**S1 Table. Proposed standard English and Dundonian non-cognate stimuli.**
(PDF)

## Acknowledgments

We wish to thank Bethany Lane for assistance with coding responses for accuracy and variety.

## Author Contributions

**Conceptualization:** Mathieu Declerck, Neil W. Kirk.

**Formal analysis:** Mathieu Declerck, Neil W. Kirk.

**Funding acquisition:** Neil W. Kirk.

**Investigation:** Mathieu Declerck, Neil W. Kirk.

**Methodology:** Mathieu Declerck.

**Project administration:** Neil W. Kirk.

**Visualization:** Neil W. Kirk.

**Writing – original draft:** Mathieu Declerck, Neil W. Kirk.

**Writing – review & editing:** Mathieu Declerck, Neil W. Kirk.

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
