## [Decision Letter · Decision Letter 0]

11 Sep 2022

PONE-D-22-14768No evidence for a mixing benefit - A Registered Report of voluntary dialect switching.PLOS ONE

Dear Dr. Kirk,

Thank you for submitting your manuscript to PLOS ONE. After careful consideration, we feel that it has merit but does not fully meet PLOS ONE’s publication criteria as it currently stands. Therefore, we invite you to submit a revised version of the manuscript that addresses the points raised during the review process.

Three external reviewers have now evaluated your submission. They have identified a number of concerns that need to be carefully addressed in a revision of the manuscript. Please respond to all the points they have raised, paying particular attention to their requests for conceptual and methodological clarifications and their suggestions for improving the contextualisation of the study.

We look forward to receiving your revised manuscript.

Kind regards,

Jamie Males

Editorial Office

PLOS ONE

Journal Requirements:

Reviewers' comments:

Reviewer's Responses to Questions

**Comments to the Author**

1. Does the manuscript adhere to the experimental procedures and analyses described in the Registered Report Protocol?

If the manuscript reports any deviations from the planned experimental procedures and analyses, those must be reasonable and adequately justified.

Reviewer #1: Yes

Reviewer #2: Yes

Reviewer #3: Yes

2. If the manuscript reports exploratory analyses or experimental procedures not outlined in the original Registered Report Protocol, are these reasonable, justified and methodologically sound?

A Registered Report may include valid exploratory analyses not previously outlined in the Registered Report Protocol, as long as they are described as such.

Reviewer #1: Yes

Reviewer #2: Yes

Reviewer #3: Yes

3. Are the conclusions supported by the data and do they address the research question presented in the Registered Report Protocol?

The manuscript must describe a technically sound piece of scientific research with data that supports the conclusions. The conclusions must be drawn appropriately based on the research question(s) outlined in the Registered Report Protocol and on the data presented.

Reviewer #1: Yes

Reviewer #2: Yes

Reviewer #3: Partly

4. Have the authors made all data underlying the findings in their manuscript fully available?

Reviewer #1: Yes

Reviewer #2: Yes

Reviewer #3: Yes

5. Is the manuscript presented in an intelligible fashion and written in standard English?

Reviewer #1: Yes

Reviewer #2: Yes

Reviewer #3: Yes

6. Review Comments to the Author

Please use the space provided to explain your answers to the questions above. (Please upload your review as an attachment if it exceeds 20,000 characters)

Reviewer #1: This manuscript describes the results of a preregistered study examining the symmetry of switch costs and the presence of a mixing benefit (or cost) in bidialectical adults during voluntary language switching. The manuscript is well-written and the research described makes an important contribution to the field. I have only a few minor suggestions for revision.

1) Can the authors provide more detail about the Dundonian Switch Rate and English Switch Rate in Table 1? Do these names indicate switches into that dialect or from that dialect? Were the percentages calculated out of the total number of valid responses or total number of valid responses in that dialect?

2) Did the number of valid Dundonian responses during the voluntary language switching paradigm differ significantly from the number of valid English responses? Since participants report using Dundonian about 26% of the time, I would expect fewer responses in Dundonian during the task, but this could also be important to consider when drawing conclusions about the symmetry of the switch costs.

3) This section of the discussion (on page 20) is difficult to follow: “However, relying on voluntary language switching does change the typical mixing cost effect observed with involuntary language switching into a non-significant effect. So, it might be that the underlying mechanism that turns a mixing cost into a mixing benefit due to bilingual voluntary language switching could also reduce the mixing cost with bidialectals to the point that it is not significant anymore when relying on voluntary language switching.” Are the authors trying to highlight that although they did not see a mixing benefit, they also did not see a mixing cost? I think the authors should attempt to reword this section a bit for clarity.

4) On page 5, the authors write that “In the current study, we set out to investigate if this is also the case for bidialectals by letting bidialectals perform in a voluntary language switching paradigm.” Since this is a research study, it seems more appropriate to say that you asked them to perform a voluntary language switching paradigm than that you let them perform a voluntary language switching paradigm.

Reviewer #2: This manuscript reports a study examining voluntary switching and mixing effects in bidialectal speakers. Switching costs were found to be comparable for the two languages (symmetrical costs) and no mixing effects were observed. The findings are compared to effects typically observed with bilingual switchers.

I enjoyed reading the manuscript. It is clear and well written and it is always a pleasure to see more research on bidialectals. I didn't (as far as I recall) review the Stage 1 of this study. I have focused my review on the results & discussion but have two minor suggestions for clarification for the intro and methods. I am not sure if it's possible to implement them in a Stage II report, so I'll leave it up to the editor to advise on this.

Abstract (last sentences discussing results);

- The way the last part of the abstract is formulated ("no mixing benefit was observed") is entirely correct but did leave me with the impression that maybe a mixing cost was observed. The discussion very nicely discusses how no mixing effect still differs from cued tasks (which usually show mixing costs) and could suggest there is a qualitative rather than a quantitive difference here. The abstract does not fully capture this, so you might want to reword it to make it clear there was no mixing cost either (and to perhaps reflect on this further like done in the discussion).

Introduction:

- This point is now actually discussed in the discussion, but I was wondering about the way symmetrical switching costs are introduced in the introduction. The introduction presents a picture of switching costs being almost always symmetrical in voluntary switching studies. I don't think that presents a fully representative picture of the literature. This is partly because there are several voluntary switching studies that show an asymmetry (some are mentioned now in footnote 1 in the discussion). The other reason is that several of the studies (refs 8-14) mentioned in the introduction as support for symmetrical voluntary switching costs looked at more balanced bilinguals who might not always (and indeed in some studies do not) show asymmetrical cued switching costs either. Again, I am not sure if you can make minor edits for clarification to the introduction at this point, but if you can, I would suggest at least mentioning that asymmetrical voluntary costs have been observed in some studies.

Methods:

- This really is a point of clarification that would be great to see made. Page 11 describes the set-up of the single-language blocks. Based on this description "Participants

first saw one single language block, followed by the four voluntary language switching blocks,

and then again one single language block in the other language variety than the first single language

block.", am I right in thinking that participants completed each single language block in only one language? In other words, half of the participants did Language A - switching - Language B and the other half Language B - switching - Language A? This would be good to clarify - if my interpretation of this sentence is correct, it differs from the approach used in other studies, including the three studies from my group mentioned in the preceding sentence saying the study was modelled after. If my interpreting is incorrect, a clearer formulation would help to clarify both languages were presented before and after the switching task.

Discussion:

- page 19 (at the end) states that "the lack of such a mixing benefit with bidialectals could be

interpreted as there being no proactive language control, or at least no proactive control as

measured by the voluntary mixing benefit effect".

This suggests that a possible interpretation could be that they use no proactive control at all (as measured through this effect) but perhaps a simpler (and maybe more justifiable) interpretation could be that there is no proactive control difference between single- and dual-language contexts?

- The discussion presents a nice and careful discussion of why effects might have emerged in the way they did and how they can be influenced by various variables. I would suggest two other points of discussion here though. First of all, I think it might be wise to make the reader aware of the relatively limited number of studies that actually look at voluntary mixing effects. Especially compared to the cued switching literature, there are not that many voluntary switching studies, and quite a few of them look at switching costs only and not at mixing effects. A lot of this work has also been done with specific types of bilinguals and language pairs (e.g., Basque/Spanish, English/Spanish, Chinese/English). With this relatively low number of studies it can be difficult to say what the "normal" patterns are. In this light, you could argue that no mixing effect differs from mixing benefits observed in previous studies but you could also argue that it is in line with some studies finding this benefit in only one language and no effect (or even a cost) in the other. My second point to make is that in general it can be quite difficult to draw conclusions about how these bidialectals might differ from bilinguals, simply because language control might differ across different types of language users (including language pairs). This is discussed on page 20, finishing with "Similarly, the results of the current study could be interpreted as bidialectals and bilinguals not relying on exactly the same language control

processes." I would suggest to discuss that this could maybe also reflect differences between different types of language pairs and/or language users rather than a (seemingly categorical) difference between bidialectals and bilinguals as such.

Angela de Bruin

Reviewer #3: This is an interesting study of voluntary switching in bidialectal speakers. There is an emerging literature involving voluntary switching paradigms, and relatively little is known about voluntary switching in bidialectals as opposed to bilinguals. The study therefore has the potential to fill a gap in the research in the field. One of the key issues I have with this is the repetition of the target items over the different blocks (see below for details). In my view this is highly problematic and may well prevent this study from being publishable in its current form.

Literature review:

Pro-active language control: Reference to Braver (2012) – the key reference to pro-active/reactive control - is missing.

There is also an increasingly substantial literature on measuring similarity between languages, which is not referenced in this study. The issue of similarity is relevant for studies of bidialectals.

See eg. Schepens, J., Van Hout, R., & Jaeger, T. F. (2020). Big data suggest strong constraints of linguistic similarity on adult language learning. Cognition, 194, 104056.

4: “This should result in a more similar L1 and L2 activation level”. Change to: more similar L1 and L2 activation levels (plural)

p. 4 “easier words are produced in the L2”. It is not clear what easier words are. Are these words with higher frequency levels? It is likely that bilinguals use their two languages for different purposes (see Grosjean’s work) and therefore particular items may be better know in the L2 than the L1 (or vice versa). If some words are “easier” in the L2, it is likely an effect of topic or a frequency effect. In the discussion section, the issue of “easy items” is also taken up. Can the authors show which items were easier among their data set? Which ones were always mentioned faster?

p. 5: The term “involuntary” is confusing. Involuntary means “done without will or conscious control”. Switching does not happen irrespective of a participant’s intentions in these trials, as they actually try to use control mechanisms. There is a cue which tells participants in which language the trial should be produced. The term used in the literature is therefore “cued switching”. Please replace throughout the text.

p. 6., line 7 Control processes…… have (not has)

p. 6, line 8: even more damning … not academic language

p.6: “the extent to which bidialectals are similar to bilinguals is still unclear”. Change to: “it is still unclear to what extent….”

p. 6: “These studies provide evidence against the claim that language control is domain general [17, 24], as the control processes within a domain (i.e.,language processing) are sometimes even different.” I would be hesitant to dismiss the claim tat language control is domain general on the basis of the studies mentioned here. There could be many different reasons for the discrepancy in findings. These are likely due to differences in methods rather than fundamental differences in underlying processes. One of the key issues, in my view, is the locus of language control (see the work of Kroll, Bobb and Wodnieczka, 2006), which the authors do not mention at all. Do the authors really want to prove that language control processes are not domain general? Does the current study provide evidence for this? I don’t think so. The key issue in the current paper is whether is a difference between bidialectals and bilinguals. Better to concentrate on the evidence for / against this claim.

p.8: Why do the authors expect symmetrical switch costs with the bidialectals? I know they refer to Gollan and Fereira to support this claim, but they do not discuss the issue of language dominance, which is likely to affect the symmetry of switch costs. If the bidialectals are dominant in one of their two languages, there should be asymmetrical switch costs. The information provided in Table 10 shows that the bidialectals in the study use Dundonian only 26% of the time, and their self-ratings of it are also lower than their self-ratings of English. In Table 2 we also see that there were more errors in the Dundonian block. So it seems participants are clearly English-dominant. The issue of dominance will therefore need to be discussed. On p.21 the authors claim participants in the current study may be similar to highly proficient bilinguals. I am not sure what this claim is based on. The data from Table 10 does not seem to support this view.

The most problematic aspect of the methodology is that a small number of items was repeated so many times over the different blocks. There were twenty items that were presented twice in each block (6 blocks in total), so participants saw these items twelve times (apart from the familiarization phase, which meant another meeting with the items). The design is probably based on De Bruin, Samuel and Dunabeita (2020), where the same number of repetitions of items is found across blocks, but there is no discussion of the effect of repetition in that publication either. Can we still claim that retrieval processes needed for items that are repeated so often represent the processes used in situations with less repetition? I am not sure… Could distractors not have been included among the trials? These would have attenuated the effect of repetition and distracted from the target items, as is common in experimental designs. The absence of asymmetrical switch costs may well be an artefact of the large number of repetitions. Participants know that the same items are going to come back and they keep both in short term memory. The absence of interactions between type and variety (for switching and mixing analyses) may also be due to the repetition of the trials. Finally,there must be some learning effects or fatigue effects of the repetition across the blocks, but these are not discussed.

Mixing analysis: if items in English are names more slowly than in Dundee Scots, could this be the result of overcoming of inhibition of the more dominant language (which is more costly than overcoming inhibition of the less dominant language)?

7. PLOS authors have the option to publish the peer review history of their article (what does this mean?). If published, this will include your full peer review and any attached files.

Reviewer #1: No

Reviewer #2: **Yes: **Angela de Bruin

Reviewer #3: **Yes: **Jeanine Treffers-Daller

---

## [Author Response · Author response to Decision Letter 0]

23 Nov 2022

Dear Editor,

We are sending you the revised version of our manuscript named “No evidence for a mixing benefit - A Registered Report of voluntary dialect switching”. We are grateful for the comments by the reviewers and incorporated them where possible. 

A detailed point-to-point response to the questions of the reviewers is provided below. Where appropriate a reference to the respective changes in the manuscript is given and the major changes are highlighted in the manuscript.

Before delving into our Responses to the comments, we would like to emphasize that this is a Stage 2 Registered Report, with our Stage 1 paper already having been peer reviewed and published. Hence, despite some comments made by reviewers pertaining to the Introduction and Method sections, we only made small changes to these previously published sections as we did not feel comfortable deviating from this core principle of this new type of paper. Doing so could potentially nullify several of the great assets of Registered Reports (for more info about Register Reports, please see these recent papers: Chambers, C. D., & Tzavella, L. (2022). The past, present and future of registered reports. Nature Human Behaviour, 6, 29-42; Henderson, E. L., & Chambers, C. D. (2022). Ten simple rules for writing a Registered Report. PLOS Computational Biology, 1810, e1010571). 

Reviewer 1

1. Can the authors provide more detail about the Dundonian Switch Rate and English Switch Rate in Table 1? Do these names indicate switches into that dialect or from that dialect? Were the percentages calculated out of the total number of valid responses or total number of valid responses in that dialect?

Reply: We have added some additional detail to Table 1 to clarify this information (page 10). 

2. Did the number of valid Dundonian responses during the voluntary language switching paradigm differ significantly from the number of valid English responses? Since participants report using Dundonian about 26% of the time, I would expect fewer responses in Dundonian during the task, but this could also be important to consider when drawing conclusions about the symmetry of the switch costs.

Reply: Items were named slightly more frequently in Dundonian than English (56% vs. 44%). However, we do not think that dialect usage in daily life would have been very influential on the (a)symmetrical switch cost pattern. A prior study with the same type of bidialectals (i.e., Dundonian-English; Kirk, Kempe, Scott-Brown, Philipp, & Declerck, 2018) showed that those bidialectals that used their dialect relatively often (53% of the time) had a similar symmetrical switch cost pattern as those bidialectals that barely used their dialect (13% of the time). Based on this study, dialect usage does not have a large impact on the symmetrical switch cost pattern.

3. This section of the discussion (on page 20) is difficult to follow: “However, relying on voluntary language switching does change the typical mixing cost effect observed with involuntary language switching into a non-significant effect. So, it might be that the underlying mechanism that turns a mixing cost into a mixing benefit due to bilingual voluntary language switching could also reduce the mixing cost with bidialectals to the point that it is not significant anymore when relying on voluntary language switching.” Are the authors trying to highlight that although they did not see a mixing benefit, they also did not see a mixing cost? I think the authors should attempt to reword this section a bit for clarity.

Reply: Here we tried to explain that the same process(es) that might turn a mixing cost pattern in a cued language switching study into a mixing benefit pattern in a voluntary language switching experiment with bilinguals could turn a mixing cost pattern in a cued dialect switching study into a null effect in a voluntary dialect switching experiment with bidialectals. We have changed this section to make this clearer (page 21).

4. On page 5, the authors write that “In the current study, we set out to investigate if this is also the case for bidialectals by letting bidialectals perform in a voluntary language switching paradigm.” Since this is a research study, it seems more appropriate to say that you asked them to perform a voluntary language switching paradigm than that you let them perform a voluntary language switching paradigm.

Reply: This has been altered (page 5).

Reviewer 2

1. Abstract (last sentences discussing results); The way the last part of the abstract is formulated ("no mixing benefit was observed") is entirely correct but did leave me with the impression that maybe a mixing cost was observed. The discussion very nicely discusses how no mixing effect still differs from cued tasks (which usually show mixing costs) and could suggest there is a qualitative rather than a quantitive difference here. The abstract does not fully capture this, so you might want to reword it to make it clear there was no mixing cost either (and to perhaps reflect on this further like done in the discussion).

Reply: We have changed the abstract now to say that “no mixing effect was observed” (page 2). 

We did not delve too much into this, apart from the last sentence, because the abstract became too long when we did.

2. Introduction: This point is now actually discussed in the Discussion, but I was wondering about the way symmetrical switching costs are introduced in the introduction. The introduction presents a picture of switching costs being almost always symmetrical in voluntary switching studies. I don't think that presents a fully representative picture of the literature. This is partly because there are several voluntary switching studies that show an asymmetry (some are mentioned now in footnote 1 in the discussion). The other reason is that several of the studies (refs 8-14) mentioned in the introduction as support for symmetrical voluntary switching costs looked at more balanced bilinguals who might not always (and indeed in some studies do not) show asymmetrical cued switching costs either. Again, I am not sure if you can make minor edits for clarification to the introduction at this point, but if you can, I would suggest at least mentioning that asymmetrical voluntary costs have been observed in some studies.

Reply: We think this is a very valid point! We originally wanted to address this issue in some way by pointing it out as a footnote in the Discussion of this stage 2 Register Report. We did this in a footnote because it seemed we were contradicting ourselves in the Introduction, which should not be changed drastically in a Stage 2 Register Report, and the Discussion. However, we agree that this is important and so we have put more emphasis on this by including it in the main text of the Discussion (page 22). 

3. Methods: This really is a point of clarification that would be great to see made. Page 11 describes the set-up of the single-language blocks. Based on this description "Participants

first saw one single language block, followed by the four voluntary language switching blocks,

and then again one single language block in the other language variety than the first single language

block.", am I right in thinking that participants completed each single language block in only one language? In other words, half of the participants did Language A - switching - Language B and the other half Language B - switching - Language A? This would be good to clarify - if my interpretation of this sentence is correct, it differs from the approach used in other studies, including the three studies from my group mentioned in the preceding sentence saying the study was modelled after. If my interpreting is incorrect, a clearer formulation would help to clarify both languages were presented before and after the switching task.

Reply: Thank you for pointing this out! You are right that this was not correct. We have changed this now (page 11).

4. page 19 (at the end) states that "the lack of such a mixing benefit with bidialectals could be

interpreted as there being no proactive language control, or at least no proactive control as

measured by the voluntary mixing benefit effect". 

This suggests that a possible interpretation could be that they use no proactive control at all (as measured through this effect) but perhaps a simpler (and maybe more justifiable) interpretation could be that there is no proactive control difference between single- and dual-language contexts?

Reply: Thank you for this suggestion! We believe that this makes more sense than that no proactive control would be implemented at all (a statement we kept, as we cannot exclude this option based on the current study), and thus included this in the revised manuscript (page 19). 

5. The discussion presents a nice and careful discussion of why effects might have emerged in the way they did and how they can be influenced by various variables. I would suggest two other points of discussion here though. First of all, I think it might be wise to make the reader aware of the relatively limited number of studies that actually look at voluntary mixing effects. Especially compared to the cued switching literature, there are not that many voluntary switching studies, and quite a few of them look at switching costs only and not at mixing effects. A lot of this work has also been done with specific types of bilinguals and language pairs (e.g., Basque/Spanish, English/Spanish, Chinese/English). With this relatively low number of studies it can be difficult to say what the "normal" patterns are. In this light, you could argue that no mixing effect differs from mixing benefits observed in previous studies but you could also argue that it is in line with some studies finding this benefit in only one language and no effect (or even a cost) in the other. My second point to make is that in general it can be quite difficult to draw conclusions about how these bidialectals might differ from bilinguals, simply because language control might differ across different types of language users (including language pairs). This is discussed on page 20, finishing with "Similarly, the results of the current study could be interpreted as bidialectals and bilinguals not relying on exactly the same language control processes." I would suggest to discuss that this could maybe also reflect differences between different types of language pairs and/or language users rather than a (seemingly categorical) difference between bidialectals and bilinguals as such.

Reply: Regarding the first comment, we have emphasized this now in the Discussion (see below for what we exactly added to the revised manuscript; page 20).

Regarding the second comment, we already touched on this topic in the Discussion of the original submission, as we point to two studies showing differences in language control between bimodal and unimodal bilinguals. To address the comment of the Reviewer more specifically, we added the following text (page 20): “However, any language mixing difference observed here might also be due to differences between the specific languages used in prior bilingual voluntary language switching studies (Catalan-Spanish, Spanish-Basque, and Spanish-English) and the language varieties used in the current study. So, more research based on different languages and language varieties should result in more conclusive evidence, especially since the few bilingual studies that investigated the voluntary mixing benefit relied on a small number of language combinations.”

Reviewer 3

1. Literature review: Pro-active language control: Reference to Braver (2012) – the key reference to pro-active/reactive control - is missing.

There is also an increasingly substantial literature on measuring similarity between languages, which is not referenced in this study. The issue of similarity is relevant for studies of bidialectals.

See eg. Schepens, J., Van Hout, R., & Jaeger, T. F. (2020). Big data suggest strong constraints of linguistic similarity on adult language learning. Cognition, 194, 104056.

Reply: In previous studies on proactive language control, we had originally included the paper by Braver (2012). However, we were often given feedback from reviewers to remove this paper, as it is not specifically related to language. Since we know from prior research that language control processes are not entirely the same as cognitive control processes, we agree with that assessment. Hence, we have stopped including the paper by Braver (2012).

Regarding papers related to language similarity, we do not disagree that to some degree language similarity could play a role when it comes to the investigation of dialects. However, since the literature on the effect of language similarity on language control is virtually nonexistent, and because we really wanted to focus on the linguistic group at hand (i.e., bidialectals), we chose to mainly discuss studies that investigated language control in bidialectals.

2. p. 4 “easier words are produced in the L2”. It is not clear what easier words are. Are these words with higher frequency levels? It is likely that bilinguals use their two languages for different purposes (see Grosjean’s work) and therefore particular items may be better know in the L2 than the L1 (or vice versa). If some words are “easier” in the L2, it is likely an effect of topic or a frequency effect. In the discussion section, the issue of “easy items” is also taken up. Can the authors show which items were easier among their data set? Which ones were always mentioned faster?

Reply: With easier, we mean words that are processed fast (across both languages), and one factor that influences this would be word frequency, but there are also other factors involved in how “easy” a word is (e.g., phonological neighborhood density, age of acquisition, affective valence, semantic neighborhood density, etc.).

There was some variability between participants which words were produced faster, and since some were mainly produced in one language, it was difficult to provide an interpretable result. In short, another study could be run to provide an answer to this question. However, since we indicate in the Introduction (which was already published before running the experiment and thus having the results, cf. Declerck & Kirk, 2022) and Discussion that the (a)symmetry of switch costs is not very informative for our research question, we decided to forgo on conducting such an experiment.

3. p. 5: The term “involuntary” is confusing. Involuntary means “done without will or conscious control”. Switching does not happen irrespective of a participant’s intentions in these trials, as they actually try to use control mechanisms. There is a cue which tells participants in which language the trial should be produced. The term used in the literature is therefore “cued switching”. Please replace throughout the text.

Reply: This is a tricky one. The reason it is tricky is that there are several ways to indicate to participants which language to use without the use of any kind of visual/auditory cue. The most prominent way is through the use of alternating language switches (e.g., L1-L1-L2-L2-L1-…; for reviews on different types of language switching, see Declerck & Koch, 2022; Declerck & Philipp, 2015). So, simply referring to “involuntary” as cued would be incorrect. 

In the end, we did not change it, as the term “involuntary” has already been established in previous studies (e.g., Sánchez, Struys, & Declerck, 2022).

4. p. 6., line 7 Control processes…… have (not has)

Reply: Thank you for pointing this out! We have now changed this accordingly (page 6).

5. p. 4: “This should result in a more similar L1 and L2 activation level”. Change to: more similar L1 and L2 activation levels (plural)

p. 6, line 8: even more damning … not academic language

p.6: “the extent to which bidialectals are similar to bilinguals is still unclear”. Change to: “it is still unclear to what extent….”

Reply: We respectfully disagree, as we do not see an issue with these phrases.

6. p. 6: “These studies provide evidence against the claim that language control is domain general [17, 24], as the control processes within a domain (i.e., language processing) are sometimes even different.” I would be hesitant to dismiss the claim tat language control is domain general on the basis of the studies mentioned here. There could be many different reasons for the discrepancy in findings. These are likely due to differences in methods rather than fundamental differences in underlying processes. One of the key issues, in my view, is the locus of language control (see the work of Kroll, Bobb and Wodnieczka, 2006), which the authors do not mention at all. Do the authors really want to prove that language control processes are not domain general? Does the current study provide evidence for this? I don’t think so. The key issue in the current paper is whether is a difference between bidialectals and bilinguals. Better to concentrate on the evidence for / against this claim.

Reply: In this paragraph, we simply wanted to indicate that it is not clear at all whether language control is entirely domain general, for which there is quite some evidence (far more than what we presented in the paper, but we focused on the most pertinent papers regarding our study). This evidence also allows for possible qualitative differences in control processes within a domain, and thus between control processes implemented during bidialectal and bilingual language production. Therefore, this is an important part of our introduction.

7. p.8: Why do the authors expect symmetrical switch costs with the bidialectals? I know they refer to Gollan and Fereira to support this claim, but they do not discuss the issue of language dominance, which is likely to affect the symmetry of switch costs. If the bidialectals are dominant in one of their two languages, there should be asymmetrical switch costs. The information provided in Table 10 shows that the bidialectals in the study use Dundonian only 26% of the time, and their self-ratings of it are also lower than their self-ratings of English. In Table 2 we also see that there were more errors in the Dundonian block. So it seems participants are clearly English-dominant. The issue of dominance will therefore need to be discussed. On p.21 the authors claim participants in the current study may be similar to highly proficient bilinguals. I am not sure what this claim is based on. The data from Table 10 does not seem to support this view.

Reply: As indicated on page 4, any language differences based on language proficiency and usage (cf. language dominance) could be abolished when using “easier” words consistently in their less dominant language (L2), which should lead to similar L1 and L2 activation. In turn, this should lead to symmetrical switch costs. 

 Furthermore, as we indicate on page 22, prior research with similar Dundonian-English bidialectals in a cued language switching task also showed symmetrical switch costs (Declerck et al., 2021; Kirk et al., 2018). This was also the case in the current study. Symmetrical switch costs during cued language switching have typically been taken as evidence that bilinguals are highly proficient in both languages (cf. Costa & Santesteban, 2004). This would even be the case even if there was a significant overall language difference, since there could be proactive language control processes implemented that result in a “reversed language dominance effect” (cf. Costa & Santesteban, 2004). However, we did not observe any significant overall language effects in the RTs, and we did not analyze the error rates because the small number of errors would not allow for reliable results. We have clarified the text now in the Discussion (page 22).

8. The most problematic aspect of the methodology is that a small number of items was repeated so many times over the different blocks. There were twenty items that were presented twice in each block (6 blocks in total), so participants saw these items twelve times (apart from the familiarization phase, which meant another meeting with the items). The design is probably based on De Bruin, Samuel and Dunabeita (2020), where the same number of repetitions of items is found across blocks, but there is no discussion of the effect of repetition in that publication either. Can we still claim that retrieval processes needed for items that are repeated so often represent the processes used in situations with less repetition? I am not sure… Could distractors not have been included among the trials? These would have attenuated the effect of repetition and distracted from the target items, as is common in experimental designs. The absence of asymmetrical switch costs may well be an artefact of the large number of repetitions. Participants know that the same items are going to come back and they keep both in short term memory. The absence of interactions between type and variety (for switching and mixing analyses) may also be due to the repetition of the trials. Finally,there must be some learning effects or fatigue effects of the repetition across the blocks, but these are not discussed.

Reply: With respect to stimulus set size and repeated stimuli in the current study, it is worth mentioning that a large percentage of language switching studies relied on a small(ish) stimulus set size (typically even smaller than the one used here). This is the case for both seminal (Costa & Santesteban, 2004; Meuter & Allport, 1999) and very recent (e.g., Contreras-Saavedra, Koch, Schuch, & Philipp, 2021; Ivanova & Hernandez, 2021; Yahya & Ceylan, 2022; Wong & Maurer, 2021; Wu & Struys, 2021) language switching studies. 

We chose this number of stimuli and number of trials based on previous research (Jevtović et al., 2020), and because our power analysis was based on it. So, while Reviewer 3 sees this as a downside, we see this as a benefit. It provides us with a clearer comparison between the data presented here and the data of previous bilingual studies, and thus allows us to draw clearer conclusions. 

 Furthermore, it should also be mentioned that while 20 items might seem like a small number to some, because this is a bidialectal paper, this means that in essence the participants give 40 unique responses (20 in Dundonian and 20 in English). Knowing the limitations of short-term memory, we do not think that participants would hold that many possible responses in their short-term memory throughout the experiment.

Regarding the fatigue/learning influences, we do not think that this had a big qualitative impact on our results. First, we are not sure what could have been learned throughout the study, other than the names of each picture in both language varieties. As we already indicated above, this would be identical to prior bilingual studies, which we wanted to mimic to some degree to make them comparable. Second, this was by no means a long study (240 trials per participant in the main experiment, which on average took about 15 minutes to finish), especially compared to other language switching studies. Both seminal papers of language switching (Costa & Santesteban, 2004; Meuter & Allport, 1999) relied on more than 1000 trials per participant (with repeated stimuli), and asymmetrical switch costs were observed in these studies. Furthermore, many recent language switching papers rely on more than 500 trials per participant (e.g., Graham & Lavric, 2021; Liu, Li, de Bruin, & He, 2021; Roembke, Philipp, & Koch, 2022; Zheng, Roelofs, & Lemhofer, 2020; Zhu, Blanco-Elorrieta, Sun, Szakay, & Sowman, 2022). Overall, we would call this a relatively short experiment.

Finally, as this is a Stage 2 Registered Report, this design was discussed and approved by reviewers as part of our published Stage 1 Registered Report (cf. Declerck & Kirk, 2022), which we did not deviate from. 

9. Mixing analysis: if items in English are names more slowly than in Dundee Scots, could this be the result of overcoming of inhibition of the more dominant language (which is more costly than overcoming inhibition of the less dominant language)?

Reply: If we had only acquired the data in the mixed language blocks (i.e., voluntary language switching data), then one could have argued along these lines (see reversed language dominance literature; for a review, see Declerck, 2020). However, the single language block data shows a similar pattern as the mixed language blocks, with slower English than Dundonian responses. Hence, it is more likely that these bidialectals were slightly more dominant in their dialect at the time of testing them. 

Interestingly, some studies have started to compare language dominance in mixed and single language block data to bypass this possible confusion, and because it allows for a more sensitive measure of the reversed language dominance effect (e.g., Declerck, Kleinman, & Gollan, 2020).

We would like to thank you and the reviewers for these helpful comments. We hope you find our revision and response to the reviewers’ comments satisfactory.

Sincerely,

Mathieu Declerck and Neil W. Kirk

---

## [Decision Letter · Decision Letter 1]

13 Dec 2022

PONE-D-22-14768R1No evidence for a mixing benefit - A Registered Report of voluntary dialect switching.PLOS ONE

Dear Dr. Kirk,

Thank you for submitting your manuscript to PLOS ONE. After careful consideration, we feel that it has merit but does not fully meet PLOS ONE’s publication criteria as it currently stands. Therefore, we invite you to submit a revised version of the manuscript that addresses the points raised during the review process. The manuscript has been re-evaluated by two reviewers, and their comments are available below. The reviewers have raised a number of minor concerns that need attention. Could you please revise the manuscript to carefully address the concerns raised? Please submit your revised manuscript by Jan 26 2023 11:59PM. If you will need more time than this to complete your revisions, please reply to this message or contact the journal office at plosone@plos.org. Please include the following items when submitting your revised manuscript:A rebuttal letter that responds to each point raised by the academic editor and reviewer(s). You should upload this letter as a separate file labeled 'Response to Reviewers'.A marked-up copy of your manuscript that highlights changes made to the original version. You should upload this as a separate file labeled 'Revised Manuscript with Track Changes'.An unmarked version of your revised paper without tracked changes. You should upload this as a separate file labeled 'Manuscript'.If applicable, we recommend that you deposit your laboratory protocols in protocols.io to enhance the reproducibility of your results. Protocols.io assigns your protocol its own identifier (DOI) so that it can be cited independently in the future. For instructions see: https://journals.plos.org/plosone/s/submission-guidelines#loc-laboratory-protocols. Additionally, PLOS ONE offers an option for publishing peer-reviewed Lab Protocol articles, which describe protocols hosted on protocols.io. Read more information on sharing protocols at https://plos.org/protocols?utm_medium=editorial-email&utm_source=authorletters&utm_campaign=protocols.

We look forward to receiving your revised manuscript.

Kind regards,

Katrien Janin

Staff Editor

PLOS ONE

Journal Requirements:

Reviewers' comments:

Reviewer's Responses to Questions

**Comments to the Author**

1. Does the manuscript adhere to the experimental procedures and analyses described in the Registered Report Protocol?

If the manuscript reports any deviations from the planned experimental procedures and analyses, those must be reasonable and adequately justified.

Reviewer #1: Yes

Reviewer #2: Yes

2. If the manuscript reports exploratory analyses or experimental procedures not outlined in the original Registered Report Protocol, are these reasonable, justified and methodologically sound?

A Registered Report may include valid exploratory analyses not previously outlined in the Registered Report Protocol, as long as they are described as such.

Reviewer #1: Yes

Reviewer #2: Yes

3. Are the conclusions supported by the data and do they address the research question presented in the Registered Report Protocol?

The manuscript must describe a technically sound piece of scientific research with data that supports the conclusions. The conclusions must be drawn appropriately based on the research question(s) outlined in the Registered Report Protocol and on the data presented.

Reviewer #1: Yes

Reviewer #2: Yes

4. Have the authors made all data underlying the findings in their manuscript fully available?

Reviewer #1: Yes

Reviewer #2: Yes

5. Is the manuscript presented in an intelligible fashion and written in standard English?

Reviewer #1: Yes

Reviewer #2: Yes

6. Review Comments to the Author

Please use the space provided to explain your answers to the questions above. (Please upload your review as an attachment if it exceeds 20,000 characters)

Reviewer #1: This manuscript describes a study that examines bidialectal voluntary language (i.e., dialect) switching during a picture-naming task. I still believe that this manuscript makes an important contribution to the field, but there are some remaining (minor) concerns that I think the authors could better address in another revision.

P6: revised “have shown” should be “have been shown”

P11: After reading reviewer 2’s comment and then looking at the revision, I am still confused about the single language blocks. From reviewer 2’s comment, it sounds like the standard design is 2 single language blocks (one in each language), followed by the switching blocks, followed by 2 single language blocks again (one in each language). If that is the standard design in previous research, the authors should clarify why they chose to present only one language block before and only one language block after the switching.

P21: My initial comment about this sentence was that it was difficult to understand, and I don’t think that the revision made it easier to understand. I would advise breaking the ideas up into smaller sentences. It may also be helpful to specifically name the “underlying mechanism that turns a mixing cost into a mixing benefit.” I think you might mean a reduction in proactive control. If so, I would advise something like this: “Some researchers have proposed that the underlying mechanism that explains the difference between mixing costs during cued bilingual language switching and mixing benefits during voluntary bilingual language switching is a difference in proactive control. In other words, cued bilingual language switching results in mixing costs because it requires more proactive control (relative to single-language use), but voluntary bilingual language switching results in mixing benefits because it requires less proactive control (relative to single-language use). For bidialectals, the amount of proactive control needed to switch languages, relative to using a single language, may be different than for bilinguals, leading to a reduction in both mixing costs and mixing benefits.”

Reviewer #2: Thank you for making these revisions and for addressing my previous comments! I'm happy for this manuscript to be accepted in its current form.

7. PLOS authors have the option to publish the peer review history of their article (what does this mean?). If published, this will include your full peer review and any attached files.

Reviewer #1: No

Reviewer #2: No

---

## [Author Response · Author response to Decision Letter 1]

19 Dec 2022

Dear Editor,

We are sending you the revised version of our manuscript named “No evidence for a mixing benefit - A Registered Report of voluntary dialect switching”. We are grateful for the comments by the reviewers and incorporated them where possible. 

A detailed point-to-point response to the questions of the reviewers is provided below. Where appropriate a reference to the respective changes in the manuscript is given and the major changes are highlighted in the manuscript.

Reviewer 1

1. P6: revised “have shown” should be “have been shown”

Reply: We have made the suggested amendment. 

2. P11: After reading reviewer 2’s comment and then looking at the revision, I am still confused about the single language blocks. From reviewer 2’s comment, it sounds like the standard design is 2 single language blocks (one in each language), followed by the switching blocks, followed by 2 single language blocks again (one in each language). If that is the standard design in previous research, the authors should clarify why they chose to present only one language block before and only one language block after the switching.

Reply: There is no standard in the language switching literature regarding block type order (for a discussion, see Declerck, 2020). Even when solely focusing on voluntary language switching, many different approaches have been used: Some voluntary language switching studies that relied on single and mixed language blocks started with a single language block of each language, followed by the mixed language block(s), and then again a single language block of each language (e.g., Jevtović et al., 2020). Another voluntary language switching study completely counterbalanced all possible block type orders across participants (Gollan & Ferreira, 2009), while another had both single language blocks consistently prior to the mixed language block (Grunden et al., 2020). One study even conducted the single language and mixed language blocks in different sessions one week apart, with the single language blocks performed in the first session and the mixed language blocks in the second session (Gross et al., 2015). While these bilingual voluntary language switching studies all had a different approach regarding the block type order, they all showed a mixing benefit. So, it is unlikely that the mixing benefit observed in voluntary language switching studies is due to a specific block type order.

3. P21: My initial comment about this sentence was that it was difficult to understand, and I don’t think that the revision made it easier to understand. I would advise breaking the ideas up into smaller sentences. It may also be helpful to specifically name the “underlying mechanism that turns a mixing cost into a mixing benefit.” I think you might mean a reduction in proactive control. If so, I would advise something like this: “Some researchers have proposed that the underlying mechanism that explains the difference between mixing costs during cued bilingual language switching and mixing benefits during voluntary bilingual language switching is a difference in proactive control. In other words, cued bilingual language switching results in mixing costs because it requires more proactive control (relative to single-language use), but voluntary bilingual language switching results in mixing benefits because it requires less proactive control (relative to single-language use). For bidialectals, the amount of proactive control needed to switch languages, relative to using a single language, may be different than for bilinguals, leading to a reduction in both mixing costs and mixing benefits.”

Reply: Unfortunately, this is not exactly what we meant. We have changed the text to further elaborate and clarify the idea we suggested in the previous version (pages 20-21). In short: We assume that more control processes are required in a bilingual involuntary setting than in a voluntary bilingual setting, leading to different mixing effect patterns (i.e., mixing benefit and mixing costs, respectively). A similar, but less pronounced, difference could probably be observed with bidialectals, leading to a mixing cost pattern in a bidialectal involuntary setting and no mixing effect in a bidialectal voluntary setting. 

Because it is unclear what the main underlying mechanism is of mixing costs (e.g., de Bruin et al., 2018; Ma et al., 2016), and because our results do not allow for any speculation regarding this underlying mechanism, we did not provide any specifics about which underlying mechanism this is attributed to.

Reviewer 2

1. Thank you for making these revisions and for addressing my previous comments! I'm happy for this manuscript to be accepted in its current form.

Reply: Thank you for the time and effort you put into your review, which has substantially elevated our manuscript!

We would like to thank you and the reviewers for these helpful comments. We hope you find our revision and response to the reviewers’ comments satisfactory.

Sincerely,

Mathieu Declerck and Neil W. Kirk

---

## [Decision Letter · Decision Letter 2]

8 Feb 2023

No evidence for a mixing benefit - A Registered Report of voluntary dialect switching.

PONE-D-22-14768R2

Dear Dr. Kirk,

We’re pleased to inform you that your manuscript has been judged scientifically suitable for publication and will be formally accepted for publication once it meets all outstanding technical requirements.

Kind regards,

Jie Wang, Ph.D.

Academic Editor

PLOS ONE

Additional Editor Comments (optional):

Both reviewers are satisfied with the revisions. Reviewer 2 pointed out a missing space on P21.

Reviewers' comments:

Reviewer's Responses to Questions

**Comments to the Author**

1. Does the manuscript adhere to the experimental procedures and analyses described in the Registered Report Protocol?

If the manuscript reports any deviations from the planned experimental procedures and analyses, those must be reasonable and adequately justified.

Reviewer #1: Yes

Reviewer #2: Yes

2. If the manuscript reports exploratory analyses or experimental procedures not outlined in the original Registered Report Protocol, are these reasonable, justified and methodologically sound?

A Registered Report may include valid exploratory analyses not previously outlined in the Registered Report Protocol, as long as they are described as such.

Reviewer #1: Yes

Reviewer #2: Yes

3. Are the conclusions supported by the data and do they address the research question presented in the Registered Report Protocol?

The manuscript must describe a technically sound piece of scientific research with data that supports the conclusions. The conclusions must be drawn appropriately based on the research question(s) outlined in the Registered Report Protocol and on the data presented.

Reviewer #1: Yes

Reviewer #2: Yes

4. Have the authors made all data underlying the findings in their manuscript fully available?

Reviewer #1: Yes

Reviewer #2: Yes

5. Is the manuscript presented in an intelligible fashion and written in standard English?

Reviewer #1: Yes

Reviewer #2: Yes

6. Review Comments to the Author

Please use the space provided to explain your answers to the questions above. (Please upload your review as an attachment if it exceeds 20,000 characters)

Reviewer #1: I am satisfied with the revisions and responses to my previous comments that the authors have provided. I believe this manuscript is now acceptable for publication.

Reviewer #2: I did not raise any concerns in my previous review and am happy with the responses to the other reviewer's feedback.

I just noticed that on page 21 a space seems to be missing between "switchingimpact".

7. PLOS authors have the option to publish the peer review history of their article (what does this mean?). If published, this will include your full peer review and any attached files.

Reviewer #1: No

Reviewer #2: No

---

## [Editor Report · Acceptance letter]

3 Mar 2023

PONE-D-22-14768R2 

No evidence for a mixing benefit - A Registered Report of voluntary dialect switching. 

Dear Dr. Kirk:

I'm pleased to inform you that your manuscript has been deemed suitable for publication in PLOS ONE. Congratulations! Your manuscript is now with our production department. 

Kind regards, 

on behalf of

Dr. Jie Wang 

Academic Editor

PLOS ONE